

# Dual-tasking effects on static and dynamic postural balance performance: a comparison between endurance and team sport athletes

Fabio Sarto[1], Giorgia Cona[2], Francesco Chiossi[2,3], Antonio Paoli[4],
Patrizia Bisiacchi[2], Elisabetta Patron[2] and Giuseppe Marcolin[4]

[1] School of Human Movement Sciences, University of Padova, Padova, Italy
[2] Department of General Psychology, University of Padova, Padova, Italy
[3] LMU Munich, Munich, Germany
[4] Department of Biomedical Sciences, University of Padova, Padova, Italy

Corresponding author
Giuseppe Marcolin,
giuseppe.marcolin@unipd.it

## ABSTRACT

In sports, postural balance control has been demonstrated to be one of the limiting factors of performance and a necessary component to achieve any sport technique. Team players (TP) must process and react to multiple external stimuli while executing at the same time the skills of the game. By contrast, endurance athletes (END) must perform the same gesture repetitively without a concurrent coordination of continuous stimuli-related actions. However, END are used to facilitate their physical performance by adopting cognitive strategies while performing their sport gesture. Therefore, we aimed to investigate static and dynamic balance performance in these two types of athletes, both in single and dual-task conditions. Nineteen END and sixteen TP underwent a static and a dynamic balance assessment on a dynamometric platform and an instrumented oscillating board, respectively. Among TP static but not dynamic postural balance performance was negatively affected by dual-tasking considering the area of the confidence ellipse ($p < 0.001$; $d = 0.52$) and the sway path mean speed ($p < 0.001$; $d = 0.93$). Conversely, END unaltered static balance performance but showed an overall improvement in the dynamic one when dual-tasking occurred. The limited human processing capacity accounted the worsening of the cognitive performance in both TP ($p < 0.05$; $d = 0.22$) and END ($p < 0.001$; $d = 0.37$). Although TP are more used coping dual tasking, the better performance of END could be accounted for by the employment of the external attentive focus (i.e. counting backward aloud) that called into play a strategy close to those adopted during training and competitions. These surprising results should be considered when driving and developing new trainings for team players in dual-tasking conditions.

## INTRODUCTION

Postural balance control (PBC) is the act of maintaining (i.e., maintenance of a posture), achieving (i.e., voluntary moving) or restoring (i.e., reacting to external disturbance) a

state of balance during any posture or activity (*Pollock et al., 2000*). In static conditions, postural performance is generally referred as the ability to reduce body sway in ordinary postures as well as the ability to retain body balance in demanding static postures (*Paillard & Noé, 2015*). In dynamic tasks, postural performance represents the ability to control the body balance during complex movements and challenging postural conditions (e.g., during external mechanical perturbations) to prevent falls (*Paillard & Noé, 2015*). PBC is a multifactorial ability, thus, it is influenced by several intrinsic factors such as age (*Ruffieux et al., 2015*), anthropometry (*Chiari, Rocchi & Cappello, 2002*; *Hue et al., 2007*), physiological and physiopathological state (*Paillard, 2017b*; *Rinalduzzi et al., 2015*), and motor experience (*Paillard, 2017a*). In sports, PBC has been demonstrated to be one of the limiting factors of performance as well as to be related to the risk of injuries (*Zemková, 2014*). Moreover, regardless of the type of sport, no sport technique is achievable without an efficient body balance (*Paillard, 2017a*). For instance, running economy (*Paillard, 2019*) or ball control in the air on one leg in soccer (*Paillard, 2017a*) are influenced by the individual level of postural skills. Likewise, a positive correlation has been shown between postural stability and performance in basketball (*Perrin, 1991*).

For a deeper insight into PBC, the dual-task (DT) paradigm (i.e., the contemporaneous performance of a second task besides the postural one) has been introduced and widely studied in different populations (*Woollacott & Shumway-Cook, 2002*). The DT paradigm assumes that the central nervous system has limited attentional resources and when multiple tasks are performed at the same time, they may lead to interference between the two tasks. If the processing capacities of the subject are exceeded, a decreased performance in one or both tasks may occur. DT has been extensively adopted in studies with older adults and pathological subjects because of their inability to effectively allocate attention to balance in multi-tasking conditions (*Lajoie et al., 1993*). Surprisingly, only a few studies have investigated the DT effects on PBC among athletes. Therefore, the comparison between team player athletes (TP) and endurance athletes (END) is of interest. Indeed, the firsts must simultaneously process multiple information executing the skills of the game (*Gabbett, Wake & Abernethy, 2011*). END must perform only continuous and cyclical repetitions of the same gesture (e.g., running, cycling, or swimming) for long distances adopting cognitive strategies and focus of attention to increase the quality of the gesture and thus the performance. On this topic, previous investigations showed that runners adopting an associative rather than a dissociative strategy ran faster (*Masters & Ogles, 1998*). But, an external focus of attention has been demonstrated to increase the running economy and the endurance performance (*Cona et al., 2015*; *Morgan, Johnson & Morgan, 1977*; *Schücker et al., 2009*). TP generally experience situations where they must perform more cognitive and/or motor tasks simultaneously (*Huang & Mercer, 2001*) and their correct execution is fundamental to succeed in the discipline. Conversely among END only concurrent cognitive tasks are employed by athletes, usually adopting dissociative strategies to increase endurance performance. Due to the intrinsic requirements of the sports described above and to the importance of PBC in both TP and END, the present study aimed to investigate static and dynamic PBC in these two categories of sport. Moreover, we deepened PBC performance, investigating if the different cognitive sport-related demands of TP and END

reflect a different postural balance performance under DT condition. We hypothesized that TP would have a better postural performance compared to END and that TP would have exhibited a smaller decrease of PBC performance than END when PBC was assessed under DT condition. This is because TP are used to deal with concurrent cognitive and/or motor tasks during training and matches while END adopted concurrent dissociative or associative cognitive strategies.

## METHODS

### Participants

Nineteen END (age: 28.32 $\pm$ 4.59 yrs, height: 1.79 $\pm$ 0.06 m, body mass: 70.21 $\pm$ 5.97 kg) and 16 TP (age: 23.44 $\pm$ 2.49 yrs, height: 1.88 $\pm$ 0.09 m, body mass: 82.66 $\pm$ 9.6 kg) took part in this study. Among TP, 7 were volleyball players, 4 basketball players, 4 field hockey players, and 1 football player. END included 11 marathon runners, 3 triathletes, and 5 ultra-marathon runners. Both TP and END trained at least 4 times per week. Inclusion criteria were the absence of acute and overuse musculoskeletal injuries, as well as ongoing neurological pathologies, vestibular, and hearing disorders. At the time of testing, all the participants were cleared for regular sport practice.

### Protocol

The tests were performed from April to July 2019 in the nutrition and exercise physiology laboratory of the Department of Biomedical Sciences, University of Padova. All participants were informed about the experimental procedures and signed an informed consent before testing. Participants performed the tests in a single session. This study was carried out in accordance with the Declaration of Helsinki. The experimental protocol was approved by the Ethics Committee of the Department of Biomedical Sciences (approval code: HEC-DSB/06-18).

### Static postural balance assessment

A bipodalic static postural balance test was performed on a force platform (AMTI BP 400600, AMTI, Watertown, USA) for SPBC assessment. During both single (ST) and DT conditions, participants were asked to stand on the platform with the arms relaxed along their sides and to gaze a target placed on a wall at 1 m distance. All trials were performed barefoot, with the heels aligned and the feet forming an angle of 30° (*Kapteyn et al., 1983*). In DT condition, we employed a counting backward task as in previous works (*Swanenburg et al., 2010*). Participants were provided a starting number above 200 and were asked to count backward by 7 aloud as accurately as possible for the duration of the whole standing trial. Two trials of 50s for each condition were recorded at a sampling rate of 100 Hz. The last 40s were considered for the analysis. The SPBC performance was assessed throughout the area of the confidence ellipse (cm$^2$) and the sway path mean speed (cm/s). The cognitive performance was evaluated with the number of correct subtractions given. For all the parameters the mean between the two trials was computed and considered for the analysis.

## Dynamic postural balance assessment

For the assessment of the DPBC, an unstable board was employed. The board enabled the oscillations only around one single axis. Two reflective markers were placed on the right edge of the board. A six-camera motion capture system (OptiTrack, NaturalPoint®, Corvallis, OR, USA) was employed to record the three-dimensional coordinates of the 2 markers and thus the motion of the board. The sampling rate was set at 100 Hz. Participants stood on the board aligning the mid-point of the platform with the mid-point of each foot (measured as half the distance between the medial malleolus and the basis of the first metatarsus), thus enabling anterior-posterior oscillations. Participants were instructed to maintain the board parallel to the floor as much as possible without moving the feet from their starting position. Before the data collection, the participants underwent a 5 min familiarization session with the board. Then, they performed two trials of 45s for each condition (i.e., ST and DT). The last 40s of each trial were considered for the analysis. The Smart Analyzer (BTS bioengineering, Milano, Italy) and Smart Tracker (BTS, bioengineering, Milano, Italy) software were used in the post-processing analysis to compute the angular oscillations of the board over the duration of the trial. The DPBC performance was assessed considering the following objective parameters as presented elsewhere (*Marcolin et al., 2016*; *Marcolin et al., 2019*): (1) the integral of the angle-time curve (Full Balance, FB), (2) the time each athlete was able to stay between $+4°$ and $-4°$ (Fine Balance, FiB) and (3) between $+8°$ and $-8°$ (Gross Balance, GB). Briefly, small values of FB correspond to a better dynamic postural performance; whilst small values of FiB and GB highlight a worse dynamic postural performance. The cognitive performance was evaluated as the number of correct subtractions given. As for the SPBC, the mean of the values collected in the 2 trials was used for the analysis.

## Statistical analysis

A Two-way ANOVA with repeated measures was carried out to detect possible effects of the type of sport (i.e., TP or END) and task condition (i.e., ST or DT) on the postural balance performance. Post hoc comparisons were corrected using the Bonferroni method. A second two-way ANOVA and Bonferroni post hoc test were performed to investigate the influence of the type of sport (i.e., TP or END) or the postural condition (i.e., SPBC and DPBC) on the cognitive performance (i.e the number of correct subtractions given). The level of significance was set at $p < 0.05$. Data analysis was performed with the software packages IBM SPSS Statistics for Windows (Version 25.0. Armonk, NY: IBM Corp). Cohen's d was calculated with G*Power 3.1.9.2 software (*Faul et al., 2007*) and evaluated as trivial ($d \geq 0.19$), small ($0.2 \leq d \leq 0.49$), medium ($0.50 \leq d \leq 0.79$) and large ($d \geq 0.80$) (*Cohen, 1992*).

## RESULTS

In the SPBC assessment, the area of the confidence ellipse was significantly influenced by the task condition ($F = 5.212$; $p < 0.05$; $\eta_p^2 = 0.136$) with a better performance (i.e., smaller area of the confidence ellipse) in the ST condition. A tendency to the statistical significance was found considering the type of sport practiced ($F = 3.605$; $p = 0.066$; $\eta_p^2$: 0.098)

with an overall better performance of END compared to TP. A statistically significant interaction was found between task and sport condition ($F = 5.230$; $p < 0.05$; $\eta_p^2 = 0.137$). Bonferroni post-hoc analysis showed an increment of the area of the confidence ellipse among TP in the DT condition compared to the ST condition ($2.89 \pm 2.65$ cm$^2$ vs $1.68 \pm 0.89$ cm$^2$; $p < 0.001$; $d = 0.52$) while END highlighted no differences comparing DT and ST conditions ($1.49 \pm 0.85$ cm$^2$ vs $1.49 \pm 0.69$ cm$^2$; $d = 0$). Moreover, END showed in the DT condition a smaller area of the confidence ellipse than TP ($1.49 \pm 0.85$ cm$^2$ vs $2.89 \pm 2.65$ cm$^2$; $p < 0.05$; $d = 0.60$). Regarding the sway path mean speed a main effect of the type of sport ($F = 4.826$; $p < 0.05$; $\eta_p^2 = 0.128$) and of the task condition ($F = 19.911$; $p < 0.001$; $\eta_p^2 = 0.376$) was detected. As for the area of the confidence ellipse, Bonferroni post hoc analysis showed an increase of the sway path mean speed in the DT condition for TP ($3.07 \pm 0.47$ cm/s vs $2.69 \pm 0.23$ cm/s; $p < 0.001$; $d = 0.93$), but not for END ($3.19 \pm 0.4$ cm/s vs $2.99 \pm 0.24$ cm/s; $d = 0.61$). Conversely, a lower sway path mean speed was detected for TP compared to END in the ST condition ($2.69 \pm 0.23$ cm/s vs $2.99 \pm 0.24$ cm/s; $p < 0.001$; $d = 1.27$) but not in DT ($3.07 \pm 0.47$ vs $3.19 \pm 0.4$; $d = 0.27$).

In the DPBC assessment (Fig. 1), a tendency to the statistical significance was detected for the task condition in the FB ($F = 3.143$; $p = 0.086$; $\eta_p^2 = 0.087$) and FiB ($F = 3.452$; $p = 0.072$; $\eta_p^2 = 0.095$). A statistically significant interaction between task condition and sport condition was detected for FiB ($F = 4.183$; $p < 0.05$; $\eta_p^2 = 0.113$) and a tendency to the statistical significance for FB ($F = 3.093$; $p = 0.088$; $\eta_p^2 = 0.086$). Post hoc analysis detected only a statistically significant increase for FiB ($p < 0.05$; $d = 0.55$) in DT for END. The same trend was observed for FB ($d = 0.48$) and GB ($d = 0.45$).

Finally, regarding the cognitive score (Fig. 2), only a main effect of postural condition ($F = 21.342$; $p < 0.001$; $\eta_p^2 = 0.393$) was observed. Post hoc analysis showed a significantly lower cognitive score in dynamic condition compared to static condition for both END ($p < 0.001$; $d = 0.37$) and TP ($p < 0.05$; $d = 0.22$).

## DISCUSSION

In the present study, we investigated the differences in SPBC and DPBC performance between TP and END. The analysis was performed both in ST and DT conditions. The main finding is that END and TP showed different behavior in coping with SPBC and DPBC both in ST and DT conditions. Considering SPBC in ST condition END showed a higher sway path speed (+11.1%) compared to TP. Whereas the area of the confidence ellipse is considered an index of the overall postural performance (the smaller the area, the better the performance (*Paillard & Noé, 2015*)), the sway path speed reflects the neuromuscular activity required to maintain balance and thus the efficiency of the postural control system (*Paillard & Noé, 2015*): the lower the speed, the better the postural efficiency. Therefore, we can assume that in ST condition END showed less efficiency (i.e., higher speed of the COP) than TP. Conversely, TP showed a worse SPBC in DT condition while END maintained their performance unchanged. These results already refuted our initial hypothesis, which was mainly based on two assumptions. The first was that DT condition has been demonstrated to simultaneously increase the complexity of the physiological and

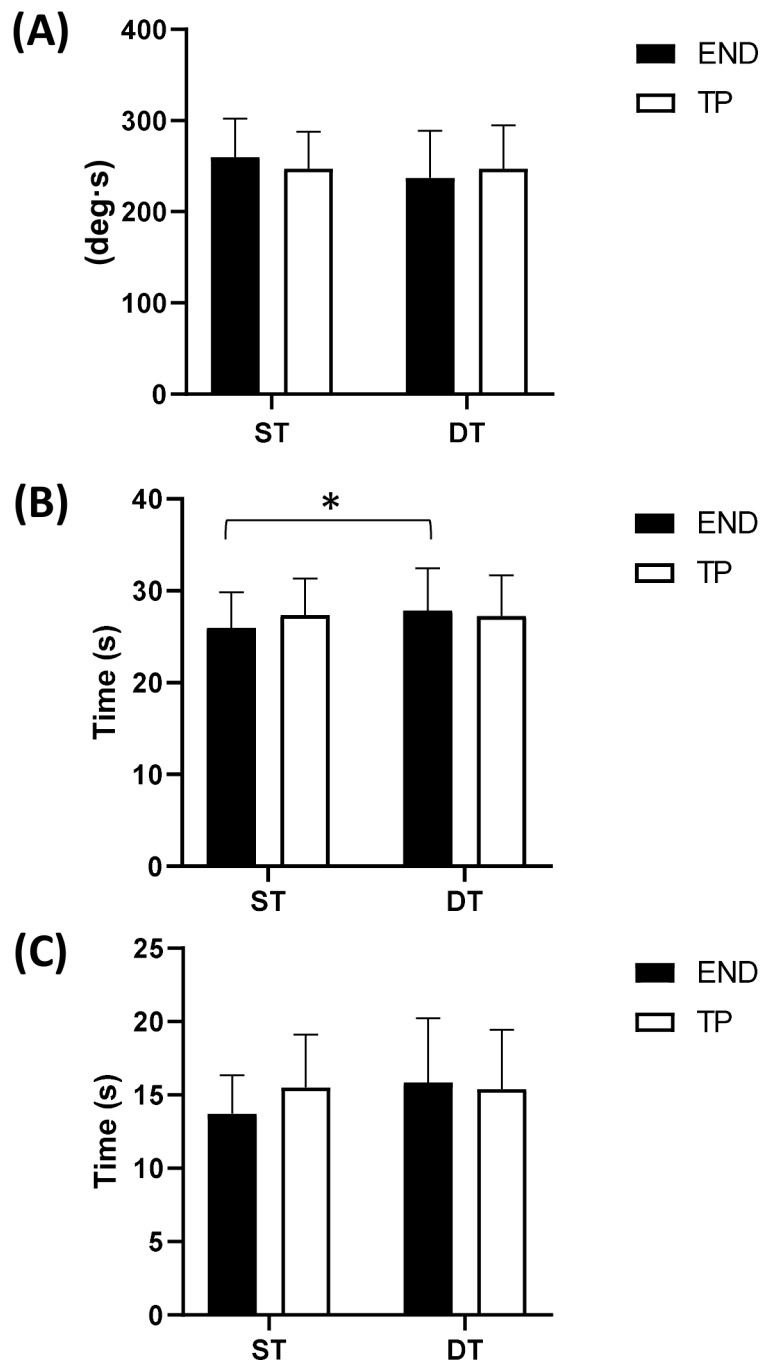

**Figure 1** **Results of the dynamic postural balance control (DPBC) parameters.** Dynamic parameters are presented as: (A) FB. (B) FiB. (C) GB. All Data are presented as mean ± standard deviation (END, endurance athletes; TP, team player athletes; ST, single-task condition; DT, dual-task condition; *, $p < 0.05$).

behavioral system leading to a cognitive-motor interference (*Ghai, Ghai & O Effenberg, 2017*; *Woollacott & Shumway-Cook, 2002*) that could have worsened the SPBC in both END and TP and not only in TP. The second was linked to the requirements of team sports where

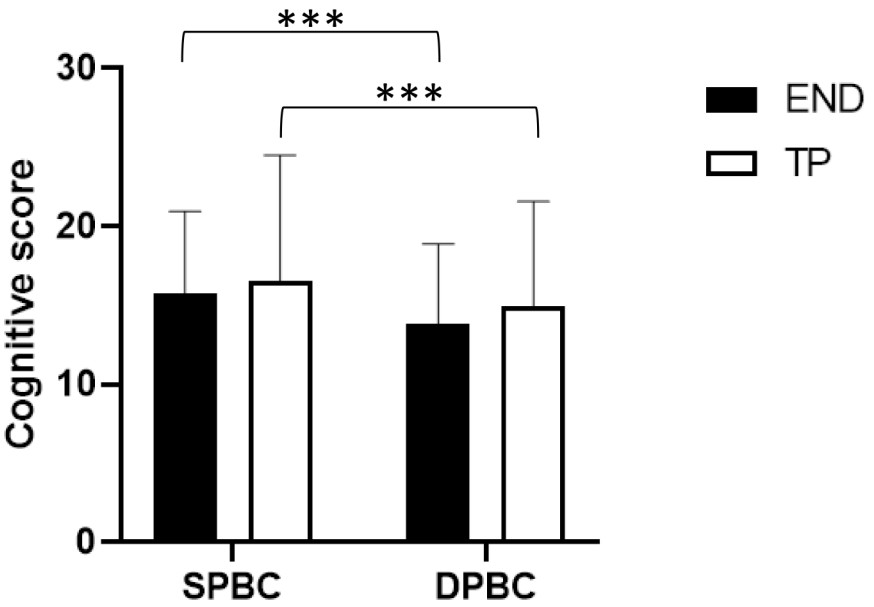

**Figure 2** **Results of the cognitive score.** All data are presented as mean ± standard deviation (END, endurance athletes; TP, team player athletes; SPBC, static postural balance control; DPBC, dynamic postural balance control; ***, $p < 0.001$).

players are used to simultaneously process multiple information while executing at the same time the skills of the game (*Gabbett, Wake & Abernethy, 2011*). Therefore, according to this assumption, we expected in DT condition a less worsening of the SPBC for TP rather than for END, due to their practice in coping with multiple information processing. However, it must be considered that SPBC mainly occurs at brainstem-spinal levels (i.e., more unconscious/automated postural strategies) due to its extremely predictable context (*Lajoie et al., 1993*). Conversely, TP always perform dynamic tasks in a less predictable context involving the cognitive processes of PBC and thus, adopting a prevalent supra-spinal postural strategy (i.e., more conscious/voluntary postural strategies) to achieve their goal-directed movements (*Lajoie et al., 1993*; *Takakusaki et al., 2017*). On the other hand, we can speculate the unaltered SPBC performance of END in the DT condition could be explained considering these athletes usually employed an external focus to enhance their endurance performance. Indeed, counting backward aloud represented an external focus that could have influenced SPBC according to the theory of reinvestment (*Masters & Maxwell, 2008*). In support to this content, it has already been shown how an external focus of attention may improve performance measures related to performance, such as running speed (*LaCaille, Masters & Heath, 2004*) and duration of the endurance task (*Morgan et al., 1983*), throughout an improvement in running economy and kinematics (*Hill et al., 2017*; *Schücker & Parrington, 2019*). In compliance with the theory of reinvestment, relatively automated motor processes (i.e., SPBC) could be impaired if they are run using a declarative memory (which requires much more attention) rather than a procedural memory (which is more automated). Therefore, the external focus (i.e., counting backward aloud) could have
reduced the conscious control of SPBC, in favor of those automated control mechanisms that effectively rule SPBC (*Takakusaki et al., 2017*).

Looking at the results of DPBC, our initial hypothesis of a better performance of TP with respect to END was also refuted. About that, it has to be considered that in DPBC the supra-spinal postural strategy is prevalent (*Lajoie et al., 1993*) with higher involvement of the cognitive process of PBC (*Takakusaki et al., 2017*) and that TP usually perform voluntary tasks in DT condition during matches and training. Therefore, we can speculate their DPBC remained unchanged when the concurrent cognitive task was introduced just because the dynamic experimental condition proposed was much closer than the static one to the on-court context. Conversely, compared to TP, END showed during DPBC a better performance as demonstrated by the statistical significance for FiB and, more marginally by the small d values for FB and GB in the DT condition. As per the SPBC, this could be explained by their consolidated habit employing an external focus during competitions to improve the performance (*Hill et al., 2017*; *Schücker & Parrington, 2019*). Indeed, counting backward aloud was an external focus among END which could have contributed to improving the DPBC performance.

Finally, the results of the cognitive performance showed no differences between TP and END, with a decrement of the correct subtractions given in the dynamic test on the oscillating board compared to the static test. The capacity sharing model (*Pashler, 1994*) can account for this worsening. This model assumes that the mental processing capacity is finite and must be shared among tasks. Therefore, since the difficulty of the cognitive task was the same both in static and dynamic conditions as well as the total processing capacity of the participants involved, we can assume that the dynamic motor task absorbed more processing capacity than the static motor task. Consequently, less processing capacity was available for the cognitive task leading to a lower number of correct subtractions given during the dynamic test.

The findings of this study have to be seen in light of some limitations. Firstly, it was not possible to match TP and END for age and anthropometry because, due to the requirements of the disciplines, TP are on average younger, higher, and heavier than END. However, our approach guaranteed the ecological validity of our study. Secondly, the standardization of the cognitive task did not allow us to find a task that was equally usual for both TP and END. A previous research (*Bergamin et al., 2014*) showed that counting backward aloud was the most challenging task that negatively influenced the center of pressure behavior. Therefore, we adopted this task, being our subjects healthy athletes. However, counting backward aloud was closer to what the END experienced in their disciplines when adopting external focus strategies rather than to what the TP experienced on-court. Indeed, in TP concurrent cognitive tasks are more sport-oriented and aimed at the success of a game scheme.

## CONCLUSIONS

The present study gave new insights on both static and dynamic postural balance control among team players and endurance athletes. Our results highlighted that the task condition

(i.e., single task or dual-task) influenced postural balance performance more than the sport practiced (i.e., team sports or endurance sports). Moreover, the most team players' habit dealing with dual tasking was not reflected in a better postural performance compared to endurance athletes. Nevertheless, these unexpected finding makes it clear that the choice of the secondary task while driving and developing training is important and should be as sport-specific as possible. Further research is needed to strengthen our hypothesis and thus to understand if, among team players, a cognitive task more sport-oriented could contribute to improving their postural balance control and consequently their sport technique.

### Funding
The authors received no funding for this work.

### Competing Interests
The authors declare there are no competing interests.

### Author Contributions
- Fabio Sarto conceived and designed the experiments, performed the experiments, analyzed the data, prepared figures and/or tables, authored or reviewed drafts of the paper, and approved the final draft.
- Giorgia Cona and Giuseppe Marcolin conceived and designed the experiments, performed the experiments, analyzed the data, authored or reviewed drafts of the paper, and approved the final draft.
- Francesco Chiossi performed the experiments, analyzed the data, prepared figures and/or tables, and approved the final draft.
- Antonio Paoli and Patrizia Bisiacchi conceived and designed the experiments, authored or reviewed drafts of the paper, and approved the final draft.
- Elisabetta Patron analyzed the data, prepared figures and/or tables, and approved the final draft.

### Human Ethics
The following information was supplied relating to ethical approvals (i.e., approving body and any reference numbers):

The experimental protocol was approved by the Ethics Committee of the Department of Biomedical Sciences, University of Padova (HEC-DSB/06-18).

### Data Availability
The raw data are available in the Supplemental File.

### Supplemental Information
Supplemental information for this article can be found online at http://dx.doi.org/10.7717/peerj.9765#supplemental-information.

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
