# Peer review of "Dual-tasking effects on static and dynamic postural balance performance: a comparison between endurance and team sport athletes"

_PeerJ, doi:10.7717/peerj.9765_

## Round 0.1 · original submission · Major Revisions

Reviewers provided favorable comments regarding the manuscript but several important points have been raised. The authors should add a conclusion at the end of the discussion section. Although the manuscript is generally well-written, authors should read more carefully to improve its readability. For example, in line 252 "In conclusion, some limitations have to be acknowledged".

Reviewer 1 ·

Basic reporting

no comment

Experimental design

For statistical analysis, Cohens'd seems a bit "tacked on" and not interpreted later on. I would add cut-offs and use it in the discussion. If not, I'd suggest getting rid of it all together.

Validity of the findings

Explain a bit further on why you want to see a lower sway velocity and a smaller sway area in your discussion. I see it in many papers, but most don't explain why that is advantageous regarding balance and postural control. Further regarding postural control, is the postural control you're measuring more unconscious or conscious and how would affect sway velocity and sway area?

·

Basic reporting

The authors state that they will 'compare static and dynamic postural balance control between TP and END athletes. Since TP and END differ for cognitive sport-related demands, they also investigated PBC performance in DT conditions'. and that is what the reported in the results, discussion, conclusion.
The article flows well and is understandable. There are some minor word usage/grammar problems. I mention some of them below my comment about figure 1. Although the current wording and sentence structure has some minor problems it is still understandable.

Fig 1 – Graphs B and C in figure 1 have the y axis labeled as ‘time (s)’. Graph A has no units assigned to the y axis.

Intro
Line 65 – For instance, high postural skills enable to run in a highly
Reviewer suggestion – need to reword to make more understandable
Line 66 - Likewise, it has been shown a positive correlation between postural stability
and performance in basketball (Perrin, 1991).
Reviewer suggestion – maybe ‘a positive correlation has been shown between…..’
Line 77 - About that, the comparison between team player athletes (TP) and endurance athletes (END) is of interest.

Reviewer suggestion – delete ‘about that’, or change in another way.

Line 78 - The firsts must simultaneously process multiple information while executing at the same time the skills of the game

Reviewer suggestion – possible delete ‘while at the same time’ (simultaneously already means at the same time)

Line 80 - The seconds must perform only continuous and cyclical repetitions
Reviewer suggestion- reword, possibly ‘END must perform…..’

Line 89 - without concurrently perform other cognitive or motor tasks

Reviewer suggestion- need to reword

Line 96 - This because TP are used to face concurrent cognitive
Reviewer suggestion – need to reword, possibly ‘This is because TP are used to dealing with concurrent cognitive tasks’

Methods

Line 127 - the sway path mean velocity

Reviewer suggestion – I realize that velocity in balance research is commonly misrepresented, BUT, velocity is a vector and since the postural sway movement that you are reporting doesn’t have a defined direction than the measure you are reporting is not velocity, but rather it is speed.

Line 148 - Briefly, the smaller is the FB, the better is the dynamic postural performance; whilst for the FiB and GB the higher is the value, the better is the dynamic postural performance.

Reviewer suggestion – need to reword

Discussion
Line 259 – Nevertheless, the unexpected finding of the present study should make think about the choice of the secondary task while driving and developing training

Reviewer suggestion – need to reword, maybe ‘Nevertheless, the unexpected finding of the present study makes it clear that the choice of the secondary task while developing training is important’,,,, or something similar

Experimental design

The experimental design is sound and resembles a similar design to many other published studies. The research question is stated at the end of the intro and matches the later sections of the article.
Methods are described appropriately.

Validity of the findings

The findings are interesting. One item that may result in questioning the results are the seeming trend of the TP to score better on the cognitive test than the END. The difference was not significant but goes in the same direction as the author's original hypothesis.
The provided data seems to be appropriate.
Conclusions match the results of the data analysis

Additional comments

This study is a pretty straight forward study on athletes, balance and dual tasking. I think it is an important contribution to the current research in that area

Reviewer 3 ·

Basic reporting

1- Authors investigated static and dynamic postural control and the effect of cognitive demand (backward counting) on postural control of team and endurance sport athletes. The manuscript grammatically is well written. The paper addressed an interesting topic and discussed multiple theories to interpret the findings. Whereas, finding causative relations in not in the scope of this research, discussing the observed results should follow an integrated mind set. However, there are some concerns that need to be addressed before deciding to publish the paper.
2- The problem statement needs more work to explicitly declare why postural control under dual task condition should be compared between two types of athletes. Endurance athletes do not need to coordinate multiple source of cognitive demands to the same extent as team players do need. So why the postural control under dual task conditions need to be compared between two groups. It need to be discussed more in the introduction.
3- The results section needs reconsideration. Some reports do not match in the "Results" and "Discussion" sections. Also some post-hoc comparisons are reported even though there were no interaction effect.

Experimental design

Overall, experimental design is adequate. But the reason for recruiting backward counting as the cognitive task is not clear, and was not the most appropriate choice in my opinion. As the authors mentioned at the end of the paper, counting backward aloud is closer to what the END experienced in their disciplines rather than to what the TP experienced on court. This could change the interpretation of the results (as the authors refer it as surprising results). I suggest small redirecting the discussion based on this fact or at least mentioning this point in the abstract.
Also aloud counting may impose additional which at least in static condition could impose some considerable sway. I suggest mentioning it as a limitation.

Validity of the findings

1- The results on DPBC are confusing and not clear. In Line 180, authors mentioned that there were no main effect or interaction on any of the variables (FB, FiB, and GB). In Lines 183-5 they reported an interaction between task condition and sport condition for FiB; and a tendency to the statistical significance for FB (which is not significant anyway). In Lines 185-7, Authors report post-hoc comparison while there are no (significant) interaction. For GB there is no tendency to significant interaction.

2- In Line 196 authors stated that "Considering SPBC in ST condition END showed a smaller area of the confidence ellipse (-11.3%) and a higher sway path velocity (+11.1%) compared to TP". This statement do not match the reported results. In the results section authors expressed that in the DT (not ST) condition END showed a smaller area of the confidence ellipse than TP. please clarify which one is correct. The statement in Lines 197-99 is not correct (exact) if the statistical results expressed in the "Results" section hold. It seems that END group holds the same SPBC in ST condition as TPs but with less efficacy (higher COP displacement velocity)...

3- In Lines 232-34 authors expressed that cognitive demand do not altered DPBC because the dynamic experimental condition proposed was much closer than the static one to the on-court context. This means that dynamic tests that subjects exposed to, has less challenge for TPs. If this holds, we should expect that TPs show better performance (smaller FB and larger FiB and GB) than ENDs. But there was no main effect of group (sport condition) on DPBC variables. Even results did not show better performance favoring TP in ST condition.

4- Authors believe that the better performance of END could be accounted by the employment of external attentive focus during training and competitions (abstract conclusion). They come into the conclusion that counting backward aloud was an external focus among ENDs which could have contributed to the better DPBC performance in ENDs (Discussion, Lines 238-9). But it is known that high-level endurance athletes (like runners, and swimmers) tend to adopt an internal focus of attention (Motor Behavior: Connecting Mind and Body for Optimal Performance, by Jeffry C Ives, P173). Also why we should not assume that TPs could employ external focus of attention too.

Additional comments

Line 29: concurrent coordination of continues stimuli-related actions...
Line 164: it is better to mention the change in variable; with a better
performance (e.g. less sway area)...
Line 197: We can assume that in ST condition...
Line 209: coping with multiple information processing...
Line 234: Conversely, compared to ST condition, ...

---

## Round 0.2 · accepted · Accept

Authors have properly addressed the reviewers´ concerns. Congratulations for meeting the high standard publications of PeerJ.

Reviewer 1 ·

Basic reporting

no comment

Experimental design

no comment

Validity of the findings

no comment

Additional comments

Thank you for addressing my concerns.

·

Basic reporting

This is the second review of this article. My first review recommended minor revisions, some of which regarded sentence structure and understandability. This iteration of the manuscript has made the changes that I suggested and is of higher quality

Experimental design

The experimental design is appropriate

Validity of the findings

The findings are well supported and well explained

Additional comments

Thank you for your time in making the suggested corrections. I think this article is a valuable contribution to the existing literature.